# Extracorporeal Removal of Thermosensitive Liposomal Doxorubicin from Systemic Circulation after Tumor Delivery to Reduce Toxicities

**DOI:** 10.3390/cancers14051322

**Published:** 2022-03-04

**Authors:** Anjan Motamarry, A. Marissa Wolfe, Krishna K. Ramajayam, Sanket Pattanaik, Thomas Benton, Yuri Peterson, Pegah Faridi, Punit Prakash, Katherine Twombley, Dieter Haemmerich

**Affiliations:** 1Department of Pediatrics, Medical University of South Carolina, Charleston, SC 29425, USA; motamarryanjan@gmail.com (A.M.); ramajaya@musc.edu (K.K.R.); twombley@musc.edu (K.T.); 2Department of Drug Discovery & Biomedical Sciences, Medical University of South Carolina, Charleston, SC 29425, USA; benton@musc.edu (T.B.); petersy@musc.edu (Y.P.); 3Wellman Center for Photomedicine, Massachusetts General Hospital, Harvard University, Boston, MA 02114, USA; 4Ralph H. Johnson VA Medical Center, Charleston, SC 29425, USA; amber.wolfe@va.gov; 5Department of Surgery, Medical University of South Carolina, Charleston, SC 29425, USA; pattanai@musc.edu; 6Department of Electrical and Computer Engineering, Kansas State University, Manhattan, KS 66506, USA; pegah@ksu.edu (P.F.); prakashp@ksu.edu (P.P.); 7Department of Bioengineering, Clemson University, Clemson, SC 29634, USA

**Keywords:** thermosensitive liposomes, hyperthermia, extracorporeal circulation

## Abstract

**Simple Summary:**

Thermosensitive liposomes (TSL) are lipid-based nanoparticles that release the encapsulated drug in response to heat. TSL are administered systemically, e.g., by intravenous infusion, and circulate within the bloodstream for an extended duration. When combined with localized heating of cancerous tumors to fever-range temperatures, TSL enable targeted delivery of chemotherapy agents, such as doxorubicin as used in this study. Following heat-activated tumor drug delivery, the majority of administered drug remains encapsulated in TSL in the bloodstream and causes undesired toxicities. We developed a method to remove this remaining drug based on an extracorporeal circuit, where first blood is heated to release drug from TSL, followed by removal of the released drug by a filter. We demonstrate removal of ~30% of the administered dose, as well as reduced drug uptake by heart tissue. The results suggest that the method can reduce toxicities that would result from remaining drug after tumor delivery.

**Abstract:**

Thermosensitive liposomal doxorubicin (TSL-Dox) combined with localized hyperthermia enables targeted drug delivery. Tumor drug uptake occurs only during hyperthermia. We developed a novel method for removal of systemic TSL-Dox remaining after hyperthermia-triggered delivery to reduce toxicities. The carotid artery and jugular vein of Norway brown rats carrying two subcutaneous BN-175 tumors were catheterized. After allowing the animals to recover, TSL-Dox was infused at 7 mg/kg dose. Drug delivery to one of the tumors was performed by inducing 15 min microwave hyperthermia (43 °C). At the end of hyperthermia, an extracorporeal circuit (ECC) comprising a heating module to release drug from TSL-Dox followed by an activated carbon filter to remove free drug was established for 1 h (*n* = 3). A computational model simulated TSL-Dox pharmacokinetics, including ECC filtration, and predicted cardiac Dox uptake. In animals receiving ECC, we were able to remove 576 ± 65 mg of Dox (29.7 ± 3.7% of the infused dose) within 1 h, with a 2.9-fold reduction of plasma AUC. Fluorescent monitoring enabled real-time quantification of blood concentration and removed drug. Computational modeling predicted that up to 59% of drug could be removed with an ideal filter, and that cardiac uptake can be reduced up to 7×. We demonstrated removal of drug remaining after tumor delivery, reduced plasma AUC, and reduced cardiac uptake, suggesting reduced toxicity.

## 1. Introduction

The majority of patients suffering from cancer are treated with some form of chemotherapy. Most chemotherapies are associated with off-target toxicities, and often those toxicities limit the chemotherapy dose that can be administered [1,2,3,4]. For doxorubicin (Dox), cardiotoxicity is the most serious type of toxicity and limits the lifetime dose of Dox that may be administered [4]. Nanoparticle drug delivery systems such as liposomes have been developed to reduce toxicities and to enhance tumor drug uptake, though still >99% of the administered drug typically does not reach the target tumor [5]. The first clinically used nanoparticle formulation was liposomal Dox (Doxil^®^) and resulted in reduced cardiotoxicity compared to an unencapsulated drug, but is associated with new toxicities such as hand-and-foot syndrome [6].

Due to the rapid distribution of most unencapsulated chemotherapy agents, removal of unencapsulated chemotherapeutics is of limited benefit as only a small fraction of the administered dose is recovered [7,8]. Additional strategies to reduce toxicities include isolated organ perfusion with drug [9] and removal of circulating liposomal encapsulated drug after adequate tumor accumulation. In the latter approach, double filtration plasmapheresis (DFPP) of liposomal drugs has been successfully employed in a small number of clinical trials [10,11,12,13]. While DFPP is effective for such long-circulating liposomes, the required duration of 4–6 h makes DFPP inadequate for liposomes with short half-lives such as thermosensitive liposomes.

Thermosensitive liposomes (TSL) are triggered nanoparticles that release the encapsulated drug in response to mild hyperthermia. Combined with localized hyperthermia, TSL enables local drug delivery. TSL-encapsulated doxorubicin (TSL-Dox) has been shown to deliver up to 20–30× more drug to tumors compared to administration of unencapsulated Dox [14]. One TSL-Dox formulation is currently in clinical trials for liver cancer and recurrent chest wall breast cancer [15,16,17]. While tumor drug uptake can be greatly enhanced with TSL-Dox, it is associated with significant toxicities that limit the dose that may be administered. In one of the clinical trials with this TSL-Dox formulation, ~40% of the patients developed mild to moderate toxicities, and the maximum tolerated dose of 50 mg/m^2^ was in the same range as for unencapsulated Dox [18].

The more recent TSL formulations are based on the intravascular triggered release paradigm, where drug release from TSL occurs while blood carrying the TSL pass through the tumor vasculature [19,20,21,22]. The released drug is then taken up by the tumor tissue. Importantly, the release and uptake of the drug take place only while hyperthermia is applied, with negligible tumor drug uptake after hyperthermia completion (Appendix A) [20,21,23]. Typically, a significant fraction of encapsulated TSL-Dox is still in circulation after hyperthermia is completed [18,24]. Therefore, intravascular triggered nanoparticles such TSL-Dox enable a new strategy to potentially reduce toxicities without impacting therapeutic efficacy by removal of this remaining TSL-Dox in circulation after tumor delivery by hyperthermia.

The goal of this study was to demonstrate this new strategy with a prototype device and remove encapsulated TSL-Dox from circulation after completion of hyperthermia based on an extracorporeal circuit (Figure 1). Such removal would reduce toxicities resulting from the leakage and uptake of TSL-Dox remaining in systemic circulation after hyperthermia. The strategy may further enable the administration of drug doses exceeding the maximum tolerated dose (MTD), since TSL-Dox is dose-limited due to those toxicities [18].

## 2. Results

### 2.1. Thermosensitive Liposome (TSL) Release Kinetics

Figure 2 shows doxorubicin release from TSL between 37 and 45 °C during the first 2.5 s, indicating that the majority of the drug was released within <2 s above 43 °C. At 37 °C, approximately 30% of the drug was released.

### 2.2. Dox Must Be Released from TSL for Effective Filtration

Prior studies have shown that activated carbon filters (ACF) can remove unencapsulated Dox effectively from blood [9,26]. In our in vitro studies, >95% of the drug could be removed from a 100 μg/mL solution of Dox in PBS (flow rate of 350μL/min; 300 mg of activated carbon) in a single pass. The filtration efficiency decreased at higher flow rates or if the amount of activated carbon was reduced (Figure 3a).

In additional in vitro studies, a solution of TSL-Dox in PBS was first pumped through the heating element for ~5 s, and then through the ACF. The heating element was set either at 37 °C or at 42 °C. Only 36.4% of Dox was removed when the heating element was set at 37 °C, whereas 89.1 ± 0.3% was removed at 42 °C (Figure 3b). The amount removed at 37 °C was the unencapsulated drug that was released at 37 °C from TSL (Figure 2). In additional studies with fluorescence-labeled TSL, we directly confirmed that liposomes were not retained by the filter (Figure 3b).

### 2.3. In vivo Removal of TSL-Dox from Systemic Blood in Extracorporeal Circuit (ECC)

Rats were injected with TSL-Dox and stratified into two groups (No ECC and ECC group). In the group without ECC, the half-life of doxorubicin was 57.3 ± 20.2 min. In the ECC group, an ECC was established 30 min after the infusion of TSL-Dox. Once ECC started, the plasma Dox concentration started to drop rapidly. We estimated that at 350 μL/min, it would require about 58 min for one blood volume (16 mL blood + 4.2 mL PBS used for priming the ECC) to pass through the filter. Dox concentration before and after the filter was quantified from serial blood samples. The amount of Dox removed (*m*_Dox_) was calculated from plasma Dox concentration before the filter (*c*_BF_), plasma Dox concentration after the filter (*c*_AF_), blood flow rate *F* (= 350 μL/min) and hematocrit (*Hct*):$$m_{Dox}=\int \;F\;\left(1-Hct\right)\;\left(c_{BF}-c_{AF}\right)\;dt$$

The hematocrit was necessary to convert blood flow to plasma flow rate. This equation was approximated by a trapezoid integration based on the concentration values (*c*_BF_, *c*_AF_) we obtained from blood samples taken every 20 min.

Within one hour of ECC (~1 whole blood volume), we were able to remove 576 ± 65 mg of Dox, which corresponded to 29.7 ± 3.7% of the infused dose (Figure 4a).

### 2.4. Pharmacokinetic Modelling

Computational pharmacokinetic models based on the same parameters (flow rate, filter efficacy) predicted that with our filter, 33.7% of the drug could be removed during 1 h of filtration and that with an ideal filter (100% efficacy), up to 59% could be removed (Figure 4b). The AUC calculated between time = 0 and infinity was 2.9× lower with filtration. In addition, the computer model predicted a significantly reduced cardiac Dox uptake as a result of the filtration, again depending on filter efficacy. With a filter of 55% efficacy, cardiac uptake was reduced by a factor of 3×, which increased up to 7× reduction with an ideal filter (Figure 4c). The predicted cardiac concentration without ECC was in the range of a recent in vivo study in rabbits (~7 μg/g) with a similar TSL-Dox formulation [27].

### 2.5. Fluorescence Monitoring Enables Real-Time In Vivo Quantification of Drug Removal

To demonstrate the ability of real-time quantification of drug removal, we performed a study inside the imaging system in one animal with fluorescence monitoring. ECC filtration was performed similarly as in the studies above, except that an imaging module was added to the extracorporeal circuit to enable real-time fluorescence monitoring of blood to measure drug concentration (Figure 5a) (in contrast, in the studies above, blood concentration was measured in extracted blood samples). In preliminary in vitro studies we demonstrated feasibility of this fluorescence monitoring approach (Appendix A). Fluorescence in vivo monitoring was performed using the imaging module, where tubes were imaged containing blood before the filter (BF), and after the filter (AF). (Figure 5b). From blood fluorescence within the tubes, Dox concentration was calculated (Figure 5c) based on a calibration curve (Appendix A). Dox concentration decreased to ~20 μg/mL within 60 min of starting the ECC. The comparison of the concentrations before and after the filter also demonstrates the diminishing filter efficacy during ECC. Filtration can be visually confirmed by looking at plasma samples before and after filtration (Figure 5d). In summary, these results demonstrate that fluorescence measurement enables the real-time monitoring of the amount of Dox in systemic circulation and of filtration efficacy.

### 2.6. Tumor Doxorubicin Uptake

Fluorescence imaging of the extracted tumors demonstrated doxorubicin uptake (Figure 6a). Drug uptake was enhanced in the tumors exposed to hyperthermia. The mean fluorescence of tumors exposed to hyperthermia was significantly higher than for unheated tumors (Figure 6b).

## 3. Discussion

Thermosensitive liposomes were first proposed about four decades ago [28]. TSL’s are triggered-release nanoparticles, which release the encapsulated agent upon exposure to mild hyperthermia. TSL are capable of carrying a variety of payloads [29]. Coupled with localized hyperthermia, TSL can deliver up to 20–30× drug to the target tumors compared to administration of unencapsulated drug [19]. For the newer TSL formulations that are based on intravascular triggered delivery [19,20,21,22], tumor drug uptake occurs only during the application of hyperthermia. Once hyperthermia is concluded, drug uptake by cancer cells ceases, and any remaining encapsulated drug does not contribute further towards tumor drug uptake (Appendix A) [20,21,23]. Since drug leakage and systemic uptake from encapsulated drugs remaining in circulation cause toxicities, the removal of this remaining drug is expected to reduce toxicities. For example, in clinical studies with a commercial TSL-Dox formulation, ~40% of the patients developed toxicities [18].

The removal of chemotherapy agents by methods such as hemofiltration or plasma filtration was employed in clinical patients (e.g., for accidental overdose), but due to the typically rapid distribution of most chemotherapy agents, only a small fraction of the administered dose can usually be recovered [7,8]. i.e., such filtration is of little benefit after administration of free drugs. Liposomal formulations, such as the liposomal Dox formulation Doxil^®^, require other filtering approaches from those used for unencapsulated drugs, since most filters (e.g., activated carbon as used here) do not retain liposomes (Figure 3b) [10]. Several clinical studies have employed double filtration plasmapheresis (DFPP) to remove the liposomal drug Doxil^®^ from systemic circulation [11,12,13]. In those prior studies, the delivery strategy was different from the current study: long-circulating liposomes were allowed to accumulate in tumors for 36–48 h after administration, and DFPP was performed subsequently [11,12]. Due to the complexity of the procedure, DFPP takes around 4–6 h to complete. DFPP is therefore not suitable for most TSL-Dox formulations which typically have a plasma half-life of only 1–2 h [18,24,30,31,32,33,34]. Unencapsulated Dox, on the other hand, can be removed very rapidly by activated carbon filters. This has been demonstrated, for example, in isolated organ perfusion studies where the liver was perfused in vivo with Dox, and ~80–90% of Dox was removed from the blood exiting the liver in a single pass through an activated carbon filter [9,26].

Based on these observations, we developed a new strategy where encapsulated TSL-Dox was removed after tumor delivery via hyperthermia (Figure 1a). An extracorporeal circuit (ECC) was established between arterial and venous catheters (Figure 1b). As the first step in the ECC, encapsulated Dox was released from TSL by exposing blood for ~5 s to 43 °C. The TSL-Dox formulation we employed released >80% of encapsulated drug within <2 s at 43 °C (Figure 2) [23]. As the second step, blood was passed through an activated carbon filter that removed the released Dox. We demonstrated in in vitro studies that the hyperthermia exposure of blood was required for effective filtration and allowed for removal of ~90% of the encapsulated drug in a single pass, while liposomes were not retained in the filter (Figure 3b). We custom-designed the heating element and filter used in this study and optimized filter parameters (activated carbon amount, flow rate) through in vitro studies (Figure 3a).

We then performed in vivo studies in a rat sarcoma model with the proposed method and demonstrated that within 60 min of filtration, ~30% of the administered dose could be recovered by filtration (Figure 4a). We did observe a diminishing filtration efficacy of our filter during the procedure, with an initial efficacy of ~80% that dropped to ~40% after 60 min (Appendix A). The average filtration efficacy during the 60 min procedure was 55%. We performed corroborating computational modeling studies where we measured the effect of filtration efficacy; these models predicted that ~34% of drug could be recovered at 55% average filtration efficacy as used in the animal studies, and 59% removal was possible with an ideal filter (100% filtration efficacy) (Figure 4b). The plasma area-under-the-curve (AUC) was reduced 2.9× with ECC filtration. In addition, the computer model predicted that cardiac drug uptake could be reduced by a factor of ~3–7×, depending on filtration efficacy (Figure 4c). Relative quantification of cardiac Dox concentration from the in vivo studies agreed with the computer model results (Figure 4d), even though the difference between groups did not reach statistical significance—likely due to the limited number of samples (3/group). This result is of clinical relevance since cardiotoxicity is the most serious type of toxicity associated with Dox exposure, and cardiotoxicity increases with cumulative Dox dose [35]. In addition, the 2.9-fold reduction in plasma AUC of Dox concentration (Figure 4a,b) suggests that other toxicities will be reduced as well since it is known that hematological toxicities such as neutropenia correlate with plasma AUC [36]. Notably, neutropenia is the dose-limiting toxicity for the commercial TSL-Dox formulation currently in clinical trials [18]. In addition to reducing systemic toxicities, the proposed method may enable the administration of doses above the maximum tolerated dose (MTD) to improve tumor drug uptake as part of the administered drug is again recovered.

Knowledge of the amount of drug removed during filtration is instrumental for any clinical application of this method since there is a lifetime dose limit for Dox, and future chemotherapy cycles are based on the dose administered in past treatments. Any drug recovered would not count towards the administered dose, and thus knowledge of the drug amount removed is of clinical relevance. While such quantification could be conducted from repeated blood samples, this approach is not practical and would also not provide real-time information. Here, we employed fluorescence monitoring to quantify drug removal in real-time. For this purpose, we developed an imaging module that enables fluorescence imaging of blood inside glass capillaries during the in vivo procedure (Figure 5b). In one animal, we performed the ECC filtration procedure inside the imaging system (Figure 5a), where the ECC included the imaging module described. Fluorescence imaging of blood after drug release from TSL-Dox via the heating element, both before and after the filter element, visualizes drug removal (Figure 5b). Based on a calibration curve (Appendix A), blood Dox concentration could be quantified from these fluorescence images (Figure 5c). While in the current study, this image analysis was performed after study completion, a dedicated system could display the blood drug concentration in real-time and could further calculate and display the total amount of drug removed during the procedure. Such information would also help to establish the optimal duration and flow rate for the filtration in a particular patient, e.g., to stop filtration once blood Dox concentration drops below a certain level. These considerations are important due to the inter-patient variability in drug pharmacokinetics. For example, the TSL-Dox formulation currently in clinical trials had inter-patient variability in plasma half-life between 0.4 and 1.8 h and varied in peak plasma Dox concentration between 17.2 and 36.5 μg/mL at the same administered dose [18]. While in the current study, we employed a dedicated fluorescence imaging system to quantify blood fluorescence, such dedicated imaging is not necessary since we only require the average blood fluorescence rather than actual images. Simpler methods for fluorescence detection are commonly available, as used for example in conventional fluorometers [37], and such methods would be more appropriate for a clinical system.

In patients, the proposed method represents a comparably simple intervention, as apart from the filtration system, only access to the patient’s circulation is necessary, for example, via vascular catheters or by vascular access ports commonly clinically used. In our rodent study, we used the carotid artery to access blood at adequate flow rates to enable rapid filtration. In human patients, venous access would provide adequate flow rates.

While activated carbon is a comparably non-specific material that removes a variety of small molecules from the blood (e.g., urea, bilirubin, etc.), there are newer materials available with more specific affinity to a particular drug [38,39,40]. Thus, the filter used in the current study could be modified or replaced with one of these newer materials to possibly improve the specificity of filtration.

The proposed method is applicable to other triggered drug delivery systems that are based on the intravascular triggered release paradigm [22]. For example, the system could be adapted for microbubbles where drug release is triggered within the extracorporeal circuit by ultrasound, i.e., replacing the heating element with an ultrasonic element that exposes blood with microbubble-encapsulated drug to ultrasound.

There are several limitations in the present study. We observed a considerable degradation of the filtration efficacy during the in vivo studies, suggesting that the filter construction would benefit from an improved design with performance similar to other commercially available filters. For example, activated carbon filters are clinically available that can extract 80–90% of Dox from the blood for 1 h at flow rates of >350 mL/min [9,26]. A disadvantage of using activated carbon is the activation of blood clotting following the adsorption of proteins, requiring the continuous infusion of heparin [41]. An alternate strategy would include specially formulated TSL with ligands that bind to specific filter material and thus would only remove the TSL together with the encapsulated drug.

We acknowledge that we have not directly shown reduced off-target toxicity. However, the fact that we recovered ~30% of administered drug associated with a 2.9-fold reduction in plasma AUC, and demonstration of reduced cardiac uptake strongly suggest that the proposed filtration will result in significantly reduced toxicities. We did not explicitly show that tumor drug uptake is not affected by the filtration, though prior studies demonstrated that no significant tumor drug uptake takes place after completion of hyperthermia (Appendix A) [20,21,23]. The proposed method may not apply to all TSL formulations since it requires rapid drug release that is present in only a few formulations. Finally, slow systemic leakage of drug from remaining TSL-Dox may contribute therapeutically to the treatment of distant cancer cells that are not targeted by hyperthermia, which would be affected by the proposed removal, presenting a potential disadvantage.

## 4. Conclusions

We present a system for the dedicated removal of TSL-encapsulated drugs from systemic circulation based on extracorporeal circulation. We demonstrated the removal of ~30% of the infused dose after the completion of tumor drug delivery by TSL-Dox, a 2.9-fold reduction of plasma-AUC, and reduced cardiac uptake by this filtration system. In addition, we demonstrated that fluorescence monitoring could quantify blood concentration and drug removal in real-time. Thus, the proposed system facilitates the removal of undelivered drugs and may thereby reduce off-target toxicities. Alternatively, the system may enable the administration of doses above the maximum tolerated dose since part of the administered dose is again removed systemically, which would enhance tumor drug uptake.

## 5. Materials and Methods

An overview of the proposed method applied in a rodent model is shown in Figure 1. Following administration of TSL-Dox and tumor hyperthermia, an extracorporeal circuit (ECC) was established between a catheterized artery and a vein. A pump passes systemic blood first through a heating element, where blood was heated to 43 °C for ~5 s to release the encapsulated drug. The employed TSL-Dox formulation releases the majority of the contained drug within <2 s at this temperature (Figure 2) [23]. In the next step, the released drug was removed by an activated carbon filter, and the filtered blood was returned to systemic circulation. In addition, fluorescence monitoring of the naturally fluorescent Dox was performed in real-time before and after the filter to quantify both the amount of Dox still in circulation as well as the amount removed by the filter. As noted, the filtration was intended to occur after tumor hyperthermia was completed, thus presumably tumor drug uptake was not reduced (Appendix A).

### 5.1. Extracorporeal Circuit (ECC) Components

#### 5.1.1. Activated Carbon Filter (ACF)

ACF for in vitro studies: 1 mL syringes (BD Biosciences, Franklin Lakes, NJ, USA) were cut into different lengths and were filled with variable amounts (100–350 mg) of activated carbon pellets (0.8 mm size, Norit steam activated, Alfa Aesar, Haverhill, MA, USA). The ends of the tubes were sealed by rubber plungers. 23-gauge needles were driven through the plungers to provide an inlet and outlet of the filter. The 0.8 mm pellets were larger than the 23-gauge needles and hence could not pass through inlet/outlet. The filters thus prepared were used for determining the filtration efficacy by pumping phosphate-buffered saline (PBS) carrying free doxorubicin (100 µg/mL) at different flow rates. In additional in vitro studies with TSL-Dox, a heating module preset at either 37 or 42 °C was used before passing the PBS carrying TSL-Dox through the filter. In a third set of in vitro studies, Rhodamine-labeled empty TSL were passed through the heating module (at either 37 or 42 °C) and through the filter at a liposomal concentration of 2 mg/mL in PBS (equivalent to the lipid concentration for TSL-Dox at 100 µg/mL Dox concentration). Rhodamine fluorescence (ex: 485 nm; em: 590 nm) was quantified by a fluorometer (Biotek Synergy HT) in samples before and after the filter to quantify removal of TSL by the filter.

ACF for in vivo studies: the ends of a 10 mL pipette (VWR Scientific, Radnor, PA, USA) was cut and filled with 2 g of activated carbon. 23 g needles were positioned at the ends, and the pipette was sealed with fast-setting epoxy (Hardman^®^, Belleville, NJ, USA) (Figure 7a). The filters were allowed to dry for 24 h and then wrapped in aluminum foil, autoclaved, and stored at room temperature until used. Before use, the filters were removed from the aluminum foil and primed with sterile PBS carrying heparin.

#### 5.1.2. Heating Element for Dox Release from TSL-Dox

For fabrication of the heating element, a Peltier module (Digi key, Inc., Thief River Falls, MN, USA, 6A input) of 4 cm × 4 cm size was glued with one side (cooled side) to an aluminum heat sink via thermally conductive epoxy resin (MG Chemicals, Surrey, BC, Canada). The surfaces were then allowed to air dry at room temperature for 2 h. Subsequently, a copper plate (4 cm × 4 cm × 0.2 cm) was glued to the second side (heated side) of the Peltier module with the thermally conductive epoxy resin, to ensure uniform surface temperature [42]. The glass capillary tubes (4 cm, 0.58 mm inner diameter, Accu-Glass, LLC, St. Louis, MO, USA) were then connected using polyurethane tubes to form a continuous flow channel. The capillary tubes were glued to the copper surface by a thin layer of thermally conductive epoxy resin and left to dry at room temperature for 24 h (Figure 7c). At a flow rate of 0.35 mL/min, the fluid (e.g., blood) spends ~5 s within the capillary tubes of the heating element, considering the inner diameter and length of the tubes. Temperature control of the Peltier element was performed using a customized proportional-integral (PI) control algorithm that controlled the voltage applied from a DC power supply to the Peltier element as described earlier [43].

#### 5.1.3. Imaging Module

We fabricated an imaging module for the measurement of fluorescence of blood before and after passing through the filter. A copper plate (3 cm × 3 cm × 0.2 cm) was coated with a thin layer of black epoxy resin, and glass microcapillaries, as used above, were placed on the resin and allowed to dry for 24 h at room temperature (Figure 7d). During experiments, fluorescence imaging was performed of the blood inside the capillaries using an imaging system (In vivo Xtreme, Bruker Corp., Billerica, MA, USA), with filters appropriate for doxorubicin (excitation 550 nm, emission 600 nm).

#### 5.1.4. Rat Back-Mount

To protect the exteriorized catheters from tampering by the animal, we designed a back-mount using 123D Design (Autodesk) and 3D-printed it using a MakerBot Replicator 5th generation (MakerBot Industries, New York, NY, USA) from biodegradable thermoplastic polylactic acid (PLA) (Figure 7b).

### 5.2. Thermosensitive Liposomal Doxorubicin (TSL-Dox)

Thermosensitive Liposomal Doxorubicin (TSL-Dox) was prepared as described earlier [24]. Briefly 1,2-dipalmitoyl-sn-glycero-3-phosphocholine (DPPC), monostearoylphosphatidylcholine (MSPC), and 1,2-distearoyl-sn-glycero-3-phosphoethanolamine-N-PEG2000 (DSPE-PEG2000) (Avanti Lipids, Alabaster, AL, USA) (DPPC:MSPC:DSPE-PEG2000 = 85.3:9.7:5.0 mol%) were dissolved in chloroform and dried under a stream of atmospheric air at room temperature for forming a thin film. The lipids were then hydrated with 300 mM citric acid buffer (pH 4.0) and extruded 5 times at 55 °C (Thermobarrel extruder; Northern Lipids, BC, Canada) through a 100 nm filter. Active loading of doxorubicin into the liposomes was carried out by pH gradient with phosphate buffered saline (PBS, pH 7.4) outside the liposomes. The release kinetics of doxorubicin from the TSL was measured between 37 and 45 °C by a millifluidic device, as described earlier [42,43].

A second batch of TSL was prepared based on the same lipid composition as above, with the addition of 0.1 mol% of Rhodamine headgroup-labeled 1,2-Dipalmitoyl-sn-glycero-3-phosphoethanolamine (Rho-DPPE) to produce empty, fluorescence-labeled TSL. This second batch was used to quantify the removal of TSL by the filter.

### 5.3. Tumor Cell Line

BN-175 cells were a gift from Dr. Timo ten Hagen (Erasmus Medical Center, Rotterdam, The Netherlands). Cells were routinely cultured at 37 °C in 5% CO_2_/95% air in Roswell Park Memorial Institute medium (RPMI) (Corning, VA, USA) supplemented with 10% inactivated fetal bovine serum (Invitrogen), 1% Penicillin, and streptomycin (Invitrogen).

### 5.4. Rat Tumor Model

Male Brown Norway rats 12–16 weeks old were purchased from Charles River Laboratories and housed in filter top polycarbonate micro isolation cages. Each cage contained 1/8″ corn cob bedding (The Andersons, Maumee, OH) with a red polycarbonate rodent house (prior to indwelling catheter implantation) or sterile 1/8″ paper strip enrichment (after surgery). Rats had ad libitum access to commercial chow (LabDiet 5V75, St. Louis, MO, USA) and bottled reverse osmosis water. The rodent housing rooms were maintained at 22 ± 1 °C on a 12:12 h light:dark cycle. All animals were acclimated for at least 5 days prior to any experimental manipulation. Specific pathogen-free status was maintained for the sendai virus, pneumonia virus of mice, sialodacryoadenitis virus, parvovirus generic NS-1 antigens, reovirus, mouse adenovirus 1 and 2, rat theilovirus, *Mycoplasma pulmonis*, as well as internal and external parasites.

Tumors in the rats were generated as previously published [44]. Briefly, 1 million BN175 sarcoma cells in 100 μL Hanks Balanced Salt Solution (HBSS) (Corning, VA, USA) were injected subcutaneously in both flanks. Tumor growth was measured by caliper measurements twice a week and tumor volume *V* was calculated using an ellipsoid approximation V = π/6 (a × b × c). When at least one of the tumors reached a volume of ~1 cm^3^, they were randomized into either the hyperthermia (HT) group (*n* = 3) or HT + ECC group (*n* = 3).

Vascular silicone catheters (3 Fr) were implanted in the jugular vein and carotid artery under anesthesia and secured by ligatures as described earlier [45]. All animal studies were approved by the Medical University of South Carolina’s Institutional Animal Care and Use Committee (IACUC).

### 5.5. Microwave Hyperthermia of Tumors

Tumor hyperthermia was performed by using a directional microwave applicator, as previously described [46]. A thermocouple was placed under the skin and above the tumor. Ultrasound gel was applied on top of the skin to ensure good dielectric contact with the microwave probe (Figure 8a). The tumor was then heated using the microwave probe to achieve a target temperature of 43 °C (measured by the thermocouple placed under the skin) for 15 min. Tumor surface temperature was also measured using an infrared camera (FLIR^®^ Systems Inc., Wilsonville, OR, USA) at the end of hyperthermia (Figure 8b).

### 5.6. In Vivo Real-Time Doxorubicin Quantification in Blood by Fluorescence Imaging

Heparinized whole blood was spiked with known quantity of doxorubicin (0–100 µg/mL). The blood samples prepared as described above were passed through the imaging module at flow rates of 350 µL/min and fluorescence imaging was performed using filters appropriate for doxorubicin (excitation 550 nm, emission 600 nm). A standard curve was established for concentrations between 0–100 µg/mL (Appendix A). During in vivo studies, blood before and after the filter was imaged and fluorescence was converted to Dox concentration based on this standard curve.

### 5.7. Doxorubicin Quantification in Plasma and PBS

Dox concentration in plasma or PBS was measured by a fluorimeter as described previously [24]. Briefly, samples were thawed. Subsequently, to 30 μL of plasma or PBS sample, 90 µL of phosphate-buffered saline and 100 µL of 10% Triton^TM^ X-100 (diluted in deionized water) were added. The fluorescence intensity of the sample was measured by a microplate reader (Synergy HT, Biotek Instruments Inc., Winooski, VT, USA) using appropriate filters for Dox (excitation 485 nm, emission 590 nm). The drug concentration was determined by a comparison against a standard curve prepared from mouse plasma samples spiked with known concentrations of Dox (1–100 µg/mL).

### 5.8. Doxorubicin Quantification in Cardiac Tissue

Doxorubicin quantification of heart tissue samples was performed by liquid-liquid extraction followed by C18-liquid chromatography selective ion recording (SIR) single quadruple mass spectrometry. Intact heart samples were homogenized in ice-cold aqueous KH_2_PO_4_ using a homogenizer (Fisher Scientific Tissuemiser, Waltham, MA, USA), resulting in a final concentration of 100 mg/mL. To 90 mL of heart homogenate, 50 mL of the 2.5 mg/mL idarubicin internal standard was added to samples. Samples were vortexed and incubated at 37 °C for 15 min. Liquid-liquid extraction was performed by adding 250 mL of acetone and 100 mL of saturated zinc sulfate. Samples were mixed by vortexing, incubated at 37 °C for 15 min, and then 200 mL were collected from the top organic layer and dried using vacuum centrifugation. Residues were reconstituted in 40:60 chromatography mobile phase A: mobile phase B.

Extracts were chromatographed using a Waters Acquity H-Class Plus UPLC equipped with a Waters Acquity UPLC BEH C18 (1.7 mm, 2.1 × 100 mm) column, stored at 30 °C. Mobile phase A comprised of milliQ water with 0.1% formic acid and mobile phase B acetonitrile with 0.1% formic acid. From the 4 °C samples manager, 4 mL of samples were injected and separated by isocratic 70% mobile phase A at 0.25 mL/min for 5 min. A Waters Acquity QDa mass spectrometer was used for the detection of analytes. Ions were generated using electrospray ionization in positive mode with the probe maintained at 600 °C, and 0.8 kV, a constant 10 V cone voltage and a source temperature of 120 °C. The molecular ions for Doxorubicin (0.40 min) and idarubicin (1.75 min) were detected using 2 separate SIR channels recording predetermined molecular ions, 543.17 and 498.17 m/z. The ratio of doxorubicin to idacubicin peak area was used for relative quantification.

### 5.9. Surgical Catheter Implantation

Under anesthesia, catheters were implanted into the jugular vein and carotid artery of Norway Brown rats. The hair on the back and neck of the animals was removed by shaving with an electric razor and hair removal cream (Nair). Under sterile surgical conditions, a 0.5 cm midline skin incision was made between the scapulae using surgical scissors. The rat was then repositioned in the dorsal position, and legs were gently restrained to each side of the table using rubber bands or tape. Two rolled sterile 4” × 4“gauze were placed under the neck to slightly hyperextend for better exposure. A 2 cm ventral cervical skin incision right of the midline of the neck at the level of the clavicle was be made using a scalpel.

#### 5.9.1. Right Jugular Vein Catheterization

Using a hemostat and blunt dissecting, the right jugular vein was separated out of the salivary and lymphatic tissues to visualize and isolate a 5 mm section of the vessel. Using a monofilament suture, a loose tie was placed on both the cranial and caudal ends of the vessel. With a microsurgical scissor, an incision large enough to pass the catheter was made. A sterile 3-french silicone catheter was inserted into the vessel with a micro dissecting hook or vein pick and forceps, and the catheter was advanced until the anchor of the catheter touched the vessel. Using the ligatures at the cranial and caudal ends, the catheter was secured to the jugular vein.

#### 5.9.2. Left Carotid Artery Catheterization

Using a hemostat and blunt dissecting, the omohyoid muscle was separated longitudinally to expose the left carotid artery and isolate a 5 mm section of the vessel. The vagus nerve was completely separated from the artery, and care was taken thus as not to damage the nerve. Using a monofilament suture, a loose tie was placed on the caudal end of the vessel, and with another tie, the cranial end of the vessel was tied off. A bulldog clamp was caudally placed above the suture to stop the blood flow following the incision. With a micro surgical scissor, an incision large enough to pass the catheter was made. The arterial catheter was inserted with the assistance of the micro dissecting hook or vessel pick and forceps. A smooth needle holder without a lock was used to hold the catheter inside the vessel, and the bulldog clamp was removed. The catheter was advanced with a pair of forceps until the anchor touched the vessel. The loose caudal ligature was tied around the catheter and vessel to secure, but not so tight as to occlude, the catheter. Using a straight hemostat and blunt dissection, a tunnel of 5 cm was created subcutaneously behind the ear and through the incision between the scapulae. The catheters were passed through the tube and the tube was removed. The ventral incision was closed with stainless steel wound clips, and the dorsal incision with monofilament sutures to secure the exteriorized catheters in place. Catheters were filled with heparinized saline and were plugged with a sterile metal plug to prevent bleeding or occlusion. A custom-designed back-mount (Figure 7b) was sterilized by autoclave and sutured to the skin surrounding the catheters using monofilament suture to protect the catheters from damage and from tampering by the animal.

#### 5.9.3. Post-Surgical Care

If the animals showed any signs of pain, animals received carpofen (2–5 mg/kg/day) subcutaneously for up to 3 days. Animals were allowed to recover for 2–3 days from surgery and then underwent the experimental procedures described below.

### 5.10. In Vivo Extracorporeal Circuit (ECC)

The patency of catheters was checked by first administering heparinized saline into the vessels, followed by withdrawal of blood into the syringe connected to the catheters. Using rubber tipped clamps, the catheters were blocked and connected to an extracorporeal circuit (ECC) consisting of sterile PVC/silicone tubing connecting the catheters to the activated carbon filter, with (for real-time fluorescence monitoring) or without the imaging module. The entire ECC line was primed with heparinized saline (10 IU/mL). The volume required for priming the ECC was ~4.2 mL. As the volume for establishing the ECC was less than 1/3rd of the calculated blood volume, we did not need any external blood donor for establishing the ECC. Once the ECC was connected to the animal, blood flow through the ECC was be maintained between 0.35 to 0.425 mL/min from the carotid artery through the heating element, filter, and imaging module (in animals where imaging was performed) and returned to the animal via the jugular vein (Figure 9). Hemofiltration was performed for up to 60 min, which corresponds to filtering approximately one blood volume. Once filtration was completed, the animals were disconnected from the ECC, and euthanasia was performed by exsanguination. The arterial line was opened for blood collection, and the tubing attached to the venous catheter was disconnected and immersed in a phosphate-buffered saline solution. The pump was maintained until breathing stopped. Once euthanasia was confirmed, the tumors were collected and rinsed in PBS, followed by fluorescence imaging of the extracted tumors.

In the current study, we accessed the blood supply between an artery and a vein (Figure 1), rather than venous access for both catheters (as is preferable in human patients). In preliminary animal studies, we were unable to sustain sufficient flow rates with venous access alone (jugular vein to tail vein). Another factor limiting the flow rate was the fragility of the glass capillary tubes used for the imaging and heating modules, which would often break due to extensive pressure when higher flow rates were used.

### 5.11. Experimental Procedure

Tumor-bearing rats were injected with TSL-Dox at a dose of 7 mg/kg under anesthesia. For animals in which ECC was performed, an ECC was established as described above (Figure 9). Following TSL-Dox administration, microwave hyperthermia was applied to one of the two tumors as described. Fluorescence imaging of both rat tumors was performed before and after hyperthermia. After hyperthermia completion, ECC filtration was initiated (if the animal was in the ECC group) for 60 min. Blood samples were obtained (~100 μL) every 20 min, both before and after the filter. In one animal, the procedure was performed inside the imaging system, and the ECC included the imaging module to enable imaging of the blood inside glass capillaries before and after the filter (Figure 5a).

### 5.12. Experimental Groups

Extracorporeal filtration without fluorescence imaging: was performed in vivo in 2 groups: (1) with extracorporeal filtration (ECC), and (2) without ECC (*n* = 3 each).

Extracorporeal filtration with fluorescence imaging: was performed in one of the animals with ECC (Note: in several animals, the experiment had to be discontinued due to the formation of blood clots inside the ECC. Presumably, the activated carbon filter removes the heparin during filtration that was administered before the procedure, which may have resulted in these blood clots).

### 5.13. Computational Modelling

We adapted a prior computer model for simulating drug delivery from TSL-Dox [20] by implementing the extracorporeal filtration circuit (ECC) (Figure 10). The model parameters were adjusted to reflect the rat model and to represent the in vivo experimental conditions (Table 1 and Appendix A). In the simulation, we assumed that blood passed through an ECC at a defined flow rate, where Dox was completely released from TSL, and a filter with a defined filtration efficacy removed the unencapsulated Dox. Any unfiltered Dox was returned to systemic circulation. In a parametric study, filtration efficacy was varied between 55% (the average efficacy of the filter used in in vivo studies) and 100%. In addition, cardiac uptake of Dox was modeled since cardiotoxicity is the most relevant toxicity for Dox. Additional details on the computational model, including equations and additional parameter values, are available in the Appendix A.

## Figures and Tables

**Figure 1 cancers-14-01322-f001:**
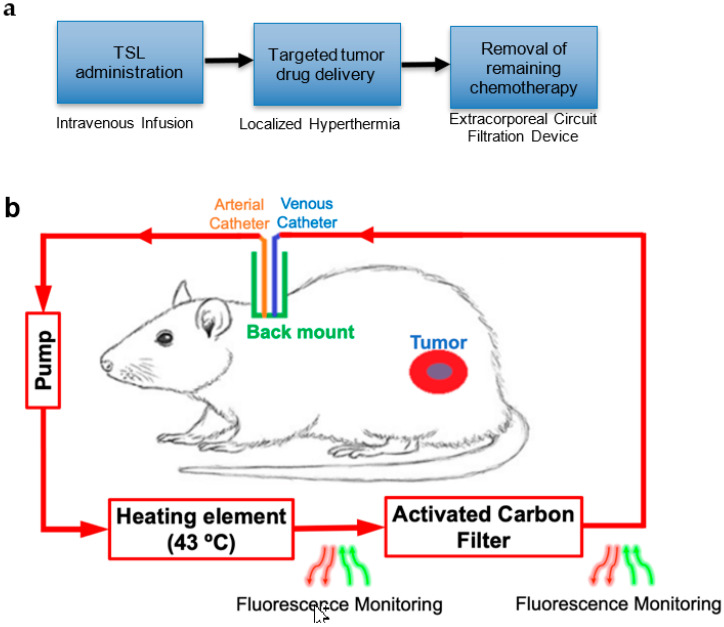
(**a**) Treatment sequence. A tumor-bearing rodent is infused with TSL-Dox, and Dox is then locally delivered to a tumor by hyperthermia. After hyperthermia, TSL-Dox remaining in circulation is removed based on a filtration system within an extracorporeal circuit (ECC). (**b**) Filtration method overview. Immediately after the conclusion of tumor hyperthermia (indicated by the red area surrounding the blue tumor), an extracorporeal circuit is established to pass blood from the carotid artery through a pump and a heating element set at 43 °C to release the drug from TSL-Dox. Then the blood is passed through an activated carbon filter (ACF) to remove the released Dox before returning the purified blood back to the systemic circulation via the jugular vein. Fluorescence monitoring of blood before and after filtration enables real-time monitoring of Dox concentration in blood. Arrows indicate the direction of blood flow. The proposed method is spatent-pending [25].

**Figure 2 cancers-14-01322-f002:**
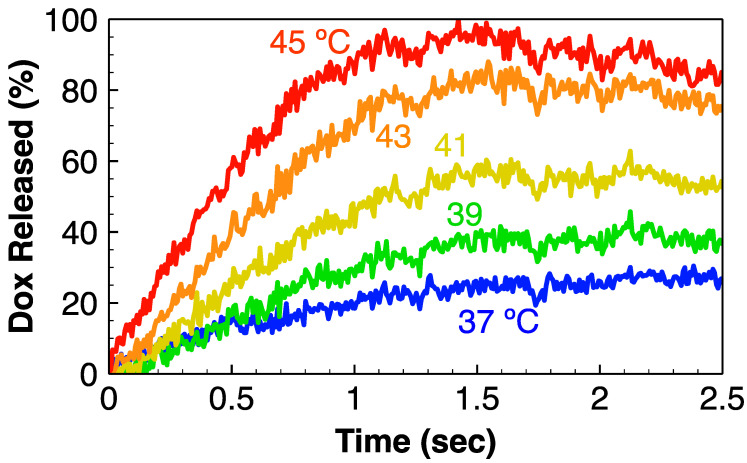
In vitro TSL release kinetics. The release was measured up to 2.5 s in phosphate-buffered saline (PBS), at 37, 39, 41, 43, and 45 °C.

**Figure 3 cancers-14-01322-f003:**
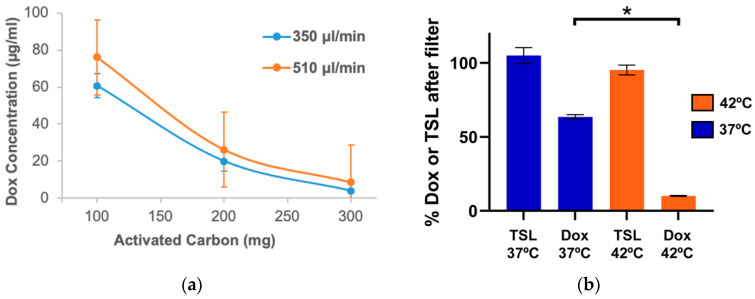
In vitro studies with an activated carbon filter. (**a**) Free Dox solution (2 mL in PBS @ 100 µg/mL concentration) was pumped through activated carbon filters (ACF) filled with 100, 200, and 300 mg activated carbon, at flow rates of 350 and 510 µL/min. The graph indicates Dox concentration after filter for the various conditions. (**b**) TSL-Dox solution (2 mL in PBS @ 100 µg/mL concentration) was passed first through the heating element set at either 37 °C or 42 °C, and then through the filter (300 mg, 350 µL/min). 36.4 ± 1.6% of Dox was removed at 37 °C, and 89.1 ± 0.3% was removed at 42 °C (*p* < 0.05). (*n* = 3 per experimental group). In addition, fluorescence-labeled empty TSL in PBS was passed through the filter after being heated to 37 °C or 42 °C. Contrary to Dox, no significant removal of TSL was observed. These results indicate that only free drug is removed by the filter. Error bars indicate standard deviation. Asterisk (*) indicates significance (*p* < 0.05).

**Figure 4 cancers-14-01322-f004:**
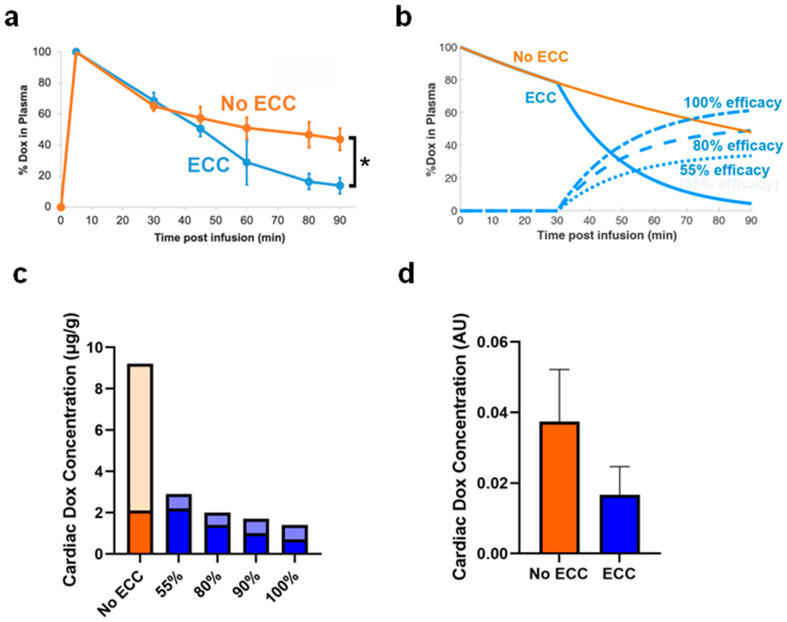
(**a**) Norway brown rats were infused with TSL-Dox (7 mg/kg) and were stratified into two groups (ECC starting at 30 min, and No ECC). In the group without ECC, the half-life of the TSL-Dox was found to be 57.3 ± 20.2 min. In the ECC group, there was a rapid drop in the concentration of the drug once the ECC was established. 29.7 ± 3.7% of the infused dose was removed from the systemic circulation due to ECC. Error bars indicate standard deviation. Asterisk (*) indicates significance (*p* < 0.05) (**b**) Computer model results showing plasma Dox concentration with, and without ECC (solid lines). In addition, % removed Dox (relative to infused dose) is shown (dashed lines), assuming filters with either 55% efficacy (similar to in vivo), 80%, and 100% efficacy. At 55% filter efficacy (average efficacy of filter used in vivo), 33.7% of the infused dose was removed. The AUC calculated between time = 0 and infinity was 40.6%*h for the ‘ECC’ case and 14.2%*h for the ‘No ECC’ case. (**c**) Computer model results showing cardiac Dox concentration of encapsulated (light) and unencapsulated/free drug (dark). Results are shown without ECC (orange) and with ECC assuming filter efficacies between 55–100%. (**d**) Relative quantification of cardiac Dox concentration measured in rat hearts from in vivo studies (*n* = 3/group). Difference in cardiac Dox uptake did not reach statistical significance (*p* = 0.13).

**Figure 5 cancers-14-01322-f005:**
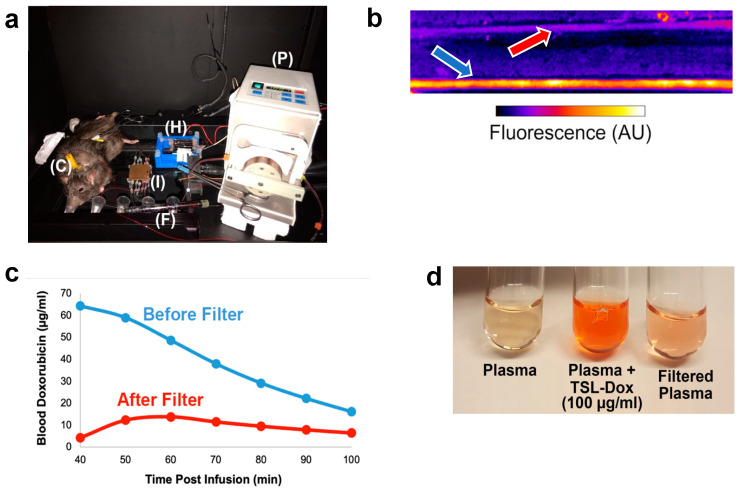
Fluorescence monitoring during filtration. (**a**) ECC set up within the imaging chamber showing rat with vascular catheters (C), heating element (H), pump (P), filter (F), and imaging module (I) containing capillary tubes (capillary tubes are at the bottom, as imaging was conducted from below). (**b**) Fluorescent imaging of capillary tubes in the imaging module during ECC shows the much higher fluorescence of blood before the filter (*blue arrow*) compared to after the filter (*red arrow*). (**c**) ECC filtration was initiated 40 min after drug infusion, and drug removal was monitored based on fluorescence imaging of blood in the capillary tubes (see **b**). (**d**) Plasma samples from the animal, before infusion of TSL-Dox (background), after infusion, and after filtration.

**Figure 6 cancers-14-01322-f006:**
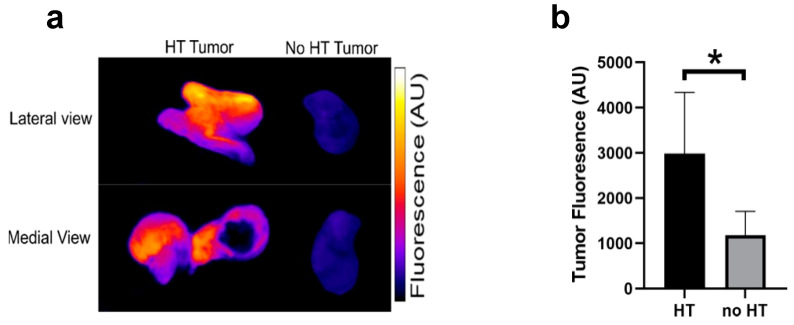
Tumor fluorescence of heated and unheated tumors. Immediately after ECC, animals were sacrificed, tumors harvested, and subjected to fluorescence imaging to visualize Dox distribution. (**a**) Enhanced tumor fluorescence of tumors exposed to hyperthermia (HT) indicates localized drug delivery from TSL-Dox. (**b**) Mean fluorescence of heated tumors was ~2.6× higher than for unheated tumors (*n* = 4/group). Asterisk (*) indicates significance (*p* < 0.05).

**Figure 7 cancers-14-01322-f007:**
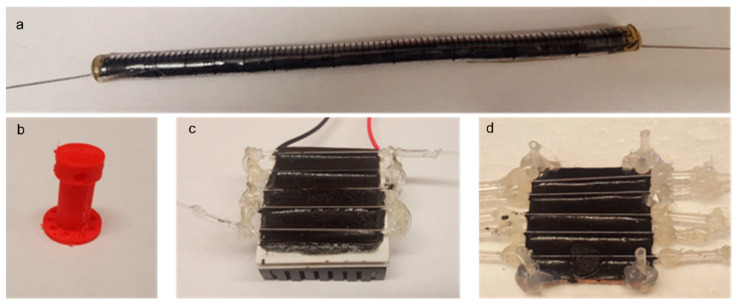
Extracorporeal circuit components. (**a**) Activated carbon filter (ACF) for in vivo studies. (**b**) a rat back-mount was 3D-printed to protect the vascular catheters from tampering by the animal. (**c**) Heating element based on a Peltier element with heat sink at the bottom, and capillary tubes at the top. (**d**) An imaging module was created by gluing capillary tubes on top a copper plate, to allow fluorescent imaging of the blood inside the tubes.

**Figure 8 cancers-14-01322-f008:**
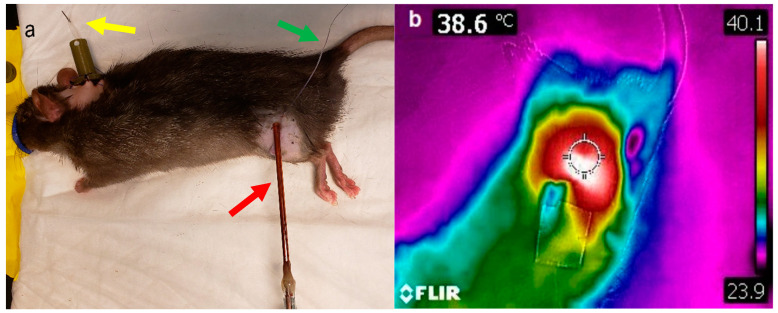
Microwave hyperthermia and infrared imaging. (**a**) Tumor bearing rats were anesthetized and the hair in the tumor area were removed by hair removal cream. TSL-Dox was infused through the venous catheter (*yellow arrow*), and the tumor was heated by a microwave probe (*red arrow*), monitored by a subcutaneous thermocouple (*green arrow*). (**b**) Hyperthermia of the tumor was further monitored by infrared thermal imaging.

**Figure 9 cancers-14-01322-f009:**
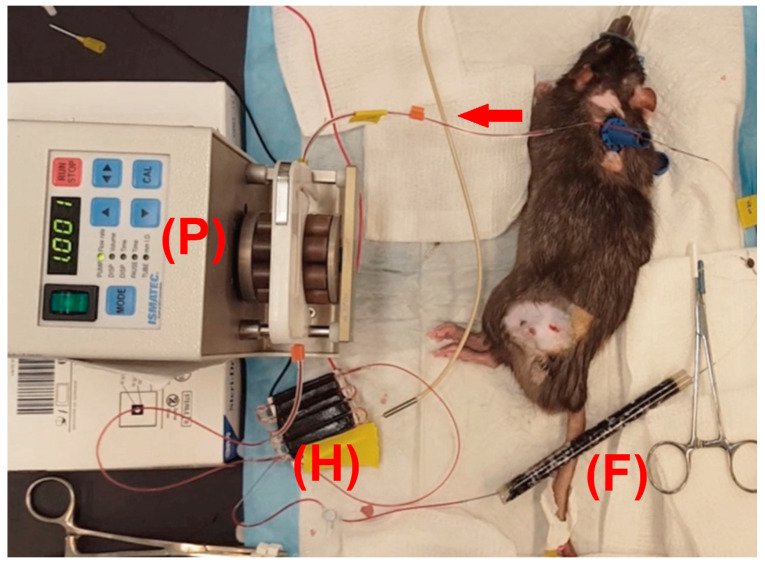
Experimental setup of ECC in rats. Blood from the carotid artery was pumped by a pump (P) through the heating element (H) set at 43 °C, then through the activated carbon filter (F), and finally returned to the animal through the jugular vein. Red arrow indicates direction of blood flow.

**Figure 10 cancers-14-01322-f010:**
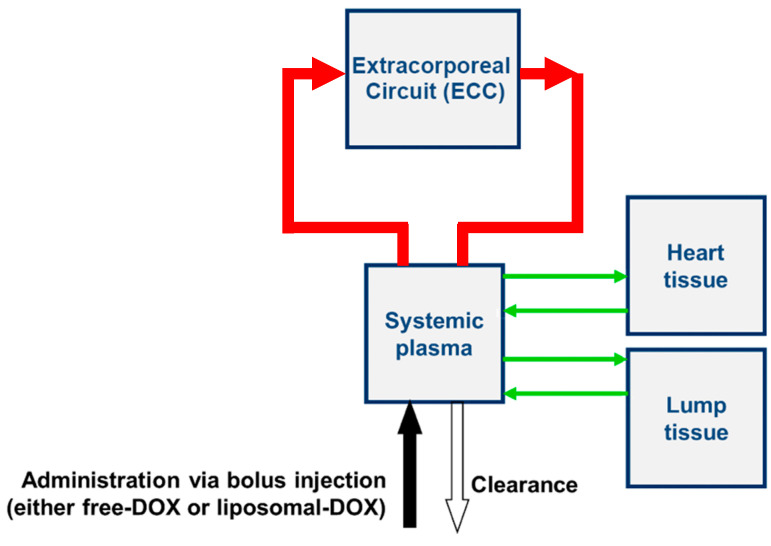
Overview of computer model. The model was adapted from a prior publication [20], with addition of the extracorporeal circuit (ECC) as indicated. For comparison to experimental results, systemic plasma Dox concentration was calculated with and without ECC filtration. In addition, Dox uptake by the heart was estimated as surrogate for cardiotoxicity.

**Table 1 cancers-14-01322-t001:** Major computer model parameters. A complete list of parameters is available in the Appendix A.

Model Parameters	Parameter Value	Comments
Weight of the Animal	250 g	
Blood volume	16 ml	Estimated from [47]
Volume of ECC (*V*_ECC_)	4.2 mL	From in vivo experiments
Dosage	7 mg/kg (=1.75 mg)	Bolus injection was assumed
Filter Perfusion Rate	0.35 mL/min	Filtration started 30 min after injection
Filtration Efficacy	55%, 80%, 100%	
Half-life of TSL-Dox	55 min	Experimentally determined

## Data Availability

The data presented in this study are available in this article and accompanying Appendix A.

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
