# Peer review of "Extracorporeal Removal of Thermosensitive Liposomal Doxorubicin from Systemic Circulation after Tumor Delivery to Reduce Toxicities"

_cancers, 2022, doi:10.3390/cancers14051322_

Round 1

Reviewer 1 Report

The manuscript is okay

Author Response

This reviewer didn't have any additional comments.

Reviewer 2 Report

Authors did a very serious revision work and replied to all my questions in very professional and satisfactory way. They have done key additional experiments that clarified one major issue I pointed out in my previous review.

I fully endorse the publication of this study.

Author Response

(The authors gave the same response as above.)

Reviewer 3 Report

Overall, the manuscript has been revised satisfactorily. Only a few points should be considered:

  1. Typographical error: Line 33- The spelling should be corrected as “thermosensisitive liposome”. Similar errors should be checked and corrected.
  2. Section 2.6 should be elaborated more, explaining the context of Fig 6a and 6b separately.

Author Response

We corrected the two issues indicated by this reviewer as described below:

Issue 1: Typographical error: Line 33- The spelling should be corrected as “thermosensisitive liposome”. Similar errors should be checked and corrected.

Response: We corrected the spelling error.

Issue 2: Section 2.6 should be elaborated more, explaining the context of Fig 6a and 6b separately.

Response: We added separate descriptions of Figure 6a and 6b to Section 2.6 as suggested.

This manuscript is a resubmission of an earlier submission. The following is a list of the peer review reports and author responses from that submission.

Round 1

Reviewer 1 Report

The study present a new method for removing cardiotoxic Dox from blood after administration in form of Dox-containing thermosensitive liposomes.
Study is well devised and done. Text is very clear. Animals used in the research have been handled according to animal care rules of the University of South Carolina.

I suggest that the manuscript can be published after the Author modify – when needed – the text by reply to my comments below (major and minor comments are reported together). Whenever possible, please comment on the manuscript, not only in the cover letter.

Comments

1) please be sure that the Abstract delivered as it was formulated for PLOS journals is acceptable also in MDPI journals

2) Lines 50-52, the Authors correctly mentioned previous approach for removal liposomes after their administration by DFPP. Probably, it is required a brief comment on the reasons why that approach is not fully satisfactory, and why a new approach is here presented. Moreover, here or in the conclusions, the two methods should be briefly compared with respect to aims, methods, results, costs, etc.

3) In Figure 2b, the result at 37 degree Celsius can be explained in two ways: either Dox is released of a certain amount at this temperature, and then adsorbed into active carbon, either that the entire liposome (with Dox inside) is adsorbed and maybe undergo rupture on carbon, Dox release, and Dox adsorption therein. What is the Author hypothesis about this result? The question can be made in another way: do liposomes interact with active carbon? A simple lipid analysis before and after passage over the active carbon can help replying this question. 

3bis) in line 180 Authors mention that liposomes are not retained by active carbon. Did the authors check that that study was referring to the same type of lipsomes used in this study? The lipid membrane composition can change the story very much, as well as the presence of PEG.

4) Line 108: maybe instead of “plasma concentration”, use “plasma Dox concentration”, etc.

5) Line 114: 575.75 +/- 65.3: I would like to warn the Authors that if the standard deviation is of the order of 65 units, reporting the average with the precision of 0.01 units is a statistical non-sense. I read this kind of numbers in several papers, but they are all wrong. This number should be best reported 575 +/- 65. But if you would like to show that your quantification method is precise at the level of 0.1 units, 575.8 +/- 65.3 is still fine (even if statistically nonsense). I will not mention significant digits issues found in the rest of the paper, but please check them accordingly.

6) If not present in the Method section, some little more explanation about the model behind Fig. 3C should be reported (see also line 210).

7) In Figure 5 caption (or before), the meaning of HT (hyperthermia) should be reported.

8) Line 165: it is not possible to mention the temperature for drug release without referring to the type of liposome used (lipid types in the membrane, etc.). The value surely depends on the chemical nature of the liposome membrane.

9) Moreover, it would be useful for the readers to know whether liposomes were stealth liposomes (peg-ylated) or not

10) In general, please check whether conclusions from other studies are directly comparable with the ones from this study, as the lipid membrane composition plays a major role in all surface-dependent phenomena.

11) Line 194: how the ‘5 s’ have been established? Is this dependent from the extracorporeal blood flow? Or does it depends on the geometry of the apparatus (lengths of the heating element)? Please explained

12) Are the results here presented limited by the heating time or by the efficiency of the active carbon adsorption capability? Please motivate your reply in the manuscript.

13) Line 214: what is the critical value of Dox in the cardiac muscle that can be tollerated? I mean, which order of magnitude? Please report values – if available.

14) Lines 257-261: Probably, heating 5s at 40 degrees Celsius is not detrimental at all for blood components. But then it came  one of my important questions to ask: what happens to all blood components while passing on active carbon filters? Do you envisage the use of materials to remove entire liposomes rather than a drug? E.g. liposomes with a ‘handle’ that can bind to some specific group on the filter resin.

15) Lines 262-265: Not clear from the text whether you refer to application of ultrasounds to release in vivo and ex vivo, or only ex vivo, … just add some words for the sake of clarity.

Author Response

Reviewer #1:

The study present a new method for removing cardiotoxic Dox from blood after administration

in form of Dox-containing thermosensitive liposomes. Study is well devised and done. Text is

very clear. Animals used in the research have been handled according to animal care rules of the

University of South Carolina.

I suggest that the manuscript can be published after the Author modify – when needed – the text

by reply to my comments below (major and minor comments are reported together). Whenever

possible, please comment on the manuscript, not only in the cover letter.

Comments

Comment #1: please be sure that the Abstract delivered as it was formulated for PLOS journals is

acceptable also in MDPI journals

Response: We removed the headings ‘Background’, ‘Methods’ etc, as suggested in the author

instructions for MDPI journals.

Comment #2: Lines 50-52, the Authors correctly mentioned previous approach for removal

liposomes after their administration by DFPP. Probably, it is required a brief comment on the

reasons why that approach is not fully satisfactory, and why a new approach is here presented.

Moreover, here or in the conclusions, the two methods should be briefly compared with respect

to aims, methods, results, costs, etc.

Response: DFPP is not adequate for thermosensitive liposomes due to the short half-life of TSLDox

of less than 2 hours – i.e. long before the 4-6 hour DFPP procedure is completed, most TSLDox

has already been cleared from systemic circulation. We added the following sentence in the

introduction: “While DFPP is effective for long circulating liposomes, the required duration of 4-

6 hours makes DFPP inadequate for liposomes with short half-lives such as thermosensitive

liposomes.” In addition, we discuss this issue in the discussion section where we added

references for various recent TSL-Dox formulations that all have plasma half-lives of 1-2 hours.

Comment #3: In Figure 2b, the result at 37 degree Celsius can be explained in two ways: either

Dox is released of a certain amount at this temperature, and then adsorbed into active carbon,

either that the entire liposome (with Dox inside) is adsorbed and maybe undergo rupture on

carbon, Dox release, and Dox adsorption therein. What is the Author hypothesis about this

result? The question can be made in another way: do liposomes interact with active carbon? A

simple lipid analysis before and after passage over the active carbon can help replying this

question.

Response: We added additional data showing the release TSL kinetics, including the release at

37 degree Celsius (new Fig. 2). These results confirm that unencapsulated doxorubicin that has

been released at 37 degree Celsius is filtered, explaining the results of prior Fig. 2b (new Fig.

3b). In addition, we performed new studies with empty TSL that were labeled with Rhodamine

to quantify liposome content before and after filtration. These new results directly demonstrate

that TSL are not removed by the filter, and only released drug is filtered (new Fig. 3b).

Comment #3b: in line 180 Authors mention that liposomes are not retained by active carbon. Did

the authors check that that study was referring to the same type of lipsomes used in this study?

The lipid membrane composition can change the story very much, as well as the presence of

PEG.

Response: As noted in our response to question #3: We performed new studies with empty TSL

that were labeled with Rhodamine, that we passed through the filter after heating the TSL to 37

or 42 ºC. The results confirm that TSL are not removed by the filter (see new Fig. 3b).

Comment #4: Line 108: maybe instead of “plasma concentration”, use “plasma Dox

concentration”, etc.

Response: We replaced “plasma concentration” with “plasma Dox concentration” in all instances

throughout the paper.

Comment #5: Line 114: 575.75 +/- 65.3: I would like to warn the Authors that if the standard

deviation is of the order of 65 units, reporting the average with the precision of 0.01 units is a

statistical non-sense. I read this kind of numbers in several papers, but they are all wrong. This

number should be best reported 575 +/- 65. But if you would like to show that your

quantification method is precise at the level of 0.1 units, 575.8 +/- 65.3 is still fine (even if

statistically nonsense). I will not mention significant digits issues found in the rest of the paper,

but please check them accordingly.

Response: We rounded these numbers to nearest integer as suggested by the reviewer.

Comment #6: If not present in the Method section, some little more explanation about the model

behind Fig. 3C should be reported (see also line 210).

Response: The computer model is described in detail in the supplementary materials.

Comment #7: In Figure 5 caption (or before), the meaning of HT (hyperthermia) should be

reported.

Response: We added the meaning of the abbreviation HT in the caption of Fig. 5 as suggested by

the reviewer.

Comment #8: Line 165: it is not possible to mention the temperature for drug release without

referring to the type of liposome used (lipid types in the membrane, etc.). The value surely

depends on the chemical nature of the liposome membrane.

Response: We removed the specific temperature (40 ºC), and only state that TSL release at mild

hyperthermic temperatures in the revised text.

Comment #9: Moreover, it would be useful for the readers to know whether liposomes were

stealth liposomes (peg-ylated) or not.

Response: Many TSL formulations are pegylated, including the one we used. We refer the reader

to review papers cited in this manuscript if they are interested in learning about pegylation of

various formulations.

Comment #10: In general, please check whether conclusions from other studies are directly

comparable with the ones from this study, as the lipid membrane composition plays a major role

in all surface-dependent phenomena.

Response: We were not sure which other studies the reviewer refers to here. However, we added

the following comment towards the end of the discussion: “The proposed method may not apply

to all TSL formulations, since it requires rapid drug release that is present in only few

formulations.”

Comment #11: Line 194: how the ‘5 s’ have been established? Is this dependent from the

extracorporeal blood flow? Or does it depends on the geometry of the apparatus (lengths of the

heating element)? Please explain.

Response: The time depends on the extracorporeal blood flow rate, as well as the diameter and

length of the tubing in the heating element. We added the following statement to ‘5.1.2. Heating

Element’ in the methods section: “At a flow rate of 0.35 mL/min, the fluid (e.g., blood) spends

~5 s within the capillary tubes of the heating element.”

Comment #12: Are the results here presented limited by the heating time or by the efficiency of

the active carbon adsorption capability? Please motivate your reply in the manuscript.

Response: The efficacy is mainly limited by the carbon adsorption capability combined with our

filter design. This is supported by the results shown in Fig. 3b and 3c. This issue is also

addressed in the discussion, where we present estimates of efficacy assuming an ideal filter. And

toward the end of the discussion we indicate the need for better filter design by stating: “We

observed a considerable degradation of the filtration efficacy during the in vivo studies,

suggesting that the filter construction would benefit from an improved design with performance

similar to other commercially available filters. For example, activated carbon filters are clinically

available that can extract 80-90% of Dox from blood for 1 h at flow rates of >350 mL/min

[9,26].”

Comment #13: Line 214: what is the critical value of Dox in the cardiac muscle that can be

tolerated? I mean, which order of magnitude? Please report values – if available.

Response: We did not find any critical value of Dox exposure in cardiac muscle from prior

studies. However, cardiotoxicity increases with administered cumulative Dox dose (Smith, MBC

Cancer 2010), which indirectly supports the notion that the proposed recapturing of administered

drug reduces cardiotoxicity. We added a statement that cardiotoxicity increases with cumulative

Dox dose in the discussion.

Comment #14: Lines 257-261: Probably, heating 5s at 40 degrees Celsius is not detrimental at all

for blood components. But then it came one of my important questions to ask: what happens to

all blood components while passing on active carbon filters? Do you envisage the use of

materials to remove entire liposomes rather than a drug? E.g. liposomes with a ‘handle’ that can

bind to some specific group on the filter resin.

Response: While activated carbon also removes other blood components, such filters have been

clinically used in human patients [9, 26], suggesting they are feasible for use in humans. The

idea of adding a ‘handle’ that can bind to a specific filter is very interesting and would be very

innovative if feasible. While this is beyond the scope of the current paper, we added the

following statement to the discussion: “An alternate strategy would include specially formulated

TSL with ligands that bind to a specific filter material, and thus would only remove the TSL

together with encapsulated drug.”

Comment #15: Lines 262-265: Not clear from the text whether you refer to application of

ultrasounds to release in vivo and ex vivo, or only ex vivo, … just add some words for the sake

of clarity.

Response: We revised the sentence as suggested: “For example, the system could be adapted for

microbubbles where drug release is triggered within the extracorporeal circuit by ultrasound, i.e.

replacing the heating element with an ultrasonic element that exposes blood with microbubbleencapsulated

drug to ultrasound.”

Reviewer 2 Report

Based on the data in Figure 3a, AUC of doxorubicin could be calculated. The difference of AUC with or without ECC may not be large as the difference in the cardiac dox concentration with or without ECC shown in figure 3c and 3d. Discussing the reason for larger reduction of doxorubicin in the heart by ECC compared to that in the systemic circulation by ECC, expressed by AUC is required.

Figure 4 b shows fluorescence imaging of capillary tubes. As the authors stated in their previous publication, reference 21, titled “Localized delivery of therapeutic doxorubicin dose across the canine blood–brain barrier with hyperthermia and temperature sensitive liposomes”, doxorubicin fluorescence is quenched while encapsulated. Therefore, the fluorescence increase in blood samples occurs only when doxorubicin released from the liposomes.  So, authors shall confirm these fluorescence signals are from free released doxorubicin. Although it is already published in reference paper 21, the characteristics of the nanoparticle, especially release kinetics shall be stated in the introduction which would help readers to understand more about this paper.

Although figure 4d clearly shows doxorubicin color in plasma sample, the blood samples the authors imaged (figure 4a and b) must be whole blood. How is the autofluorescence of red blood cells which may affect fluorescence imaging of doxorubicin in the blood? The authors shall show the standard curve of fluorescence intensity by measuring the whole blood samples spiked with known amount of free doxorubicin.

Figure 4 is most important result based on the purpose of this paper. But there was only one animal tested for ECC filtration as stated in line 137. From the statistical point of view, one animal is not enough to present even it is for the proof of concept research.

Figure 5 is not explained enough. No quantified data or the number of measurements is shown. If it is based on the imaging only one animal, the imaging shall be repeated to get some statistical comparison.

Author Response

Reviewer #2:

Comment #1: Based on the data in Figure 3a, AUC of doxorubicin could be calculated. The

difference of AUC with or without ECC may not be large as the difference in the cardiac dox

concentration with or without ECC shown in figure 3c and 3d. Discussing the reason for larger

reduction of doxorubicin in the heart by ECC compared to that in the systemic circulation by

ECC, expressed by AUC is required.

Response: We agree with the reviewer that the AUC calculated based on Fig. 3a (new Fig. 4a)

would not be that large when calculated from 0-90 min. However, AUC’s based on

pharmacokinetic data are usually calculated until infinity or until plasma concentration is close to

zero (e.g., in Fig. 4a Dox exposure continues beyond 90 min, until Dox plasma level is close to

zero). While this AUC calculation is not possible from the data in Fig. 3a (new Fig. 4a), we

performed this calculation based on our pharmacokinetic model (old Fig. 3b/new Fig. 4b), where

we could easily extend the duration beyond 90 min. We obtained an AUC=40.6 %*h for the ‘no

ECC’ case, and AUC=14.2 %*h for the ‘ECC’ case, close to three times lower. Thus, the AUC

data is in approximate agreement with the data on cardiac drug uptake. We added this AUC

calculation as additional results.

Comment #2: Figure 4 b shows fluorescence imaging of capillary tubes. As the authors stated in

their previous publication, reference 21, titled “Localized delivery of therapeutic doxorubicin

dose across the canine blood–brain barrier with hyperthermia and temperature sensitive

liposomes”, doxorubicin fluorescence is quenched while encapsulated. Therefore, the

fluorescence increase in blood samples occurs only when doxorubicin released from the

liposomes. So, authors shall confirm these fluorescence signals are from free released

doxorubicin.

Response: In additional studies for this revision we demonstrated that only free drug is removed

by the filter (new Fig. 3b). Figure 5b (old Fig. 4b) shows negligible fluorescence of blood after

passing through the filter, compared to before filtration. In combination, these data demonstrate

that fluorescence is from free (released) doxorubicin, since after filtration removes this free

doxorubicin, fluorescence is diminished. Note further, that most of the Dox is released after

heating TSL-Dox for 5 seconds (new Fig. 2). This provides additional evidence that the

fluorescence signal is in fact from free Dox, since Fig. 4b (new Fig. 5b) shows tubes with blood

that passed through the heating element.

Comment #2b: Although it is already published in reference paper 21, the characteristics of the

nanoparticle, especially release kinetics shall be stated in the introduction which would help

readers to understand more about this paper.

Response: We added a graph showing the release kinetics to address this issue (Fig. 2).

Comment #3: Although figure 4d clearly shows doxorubicin color in plasma sample, the blood

samples the authors imaged (figure 4a and b) must be whole blood. How is the autofluorescence

of red blood cells which may affect fluorescence imaging of doxorubicin in the blood? The

authors shall show the standard curve of fluorescence intensity by measuring the whole blood

samples spiked with known amount of free doxorubicin.

Response: We added the standard curve used to determine Dox concentration in whole blood

(new Fig. S4). While it is true that there is some background fluorescence present, Dox

fluorescence was sufficiently strong to allow calculation of Dox concentration at magnitudes

within our data range (~10-70 ug/ml, see new Fig. 4c).

Comment #4: Figure 4 is most important result based on the purpose of this paper. But there was

only one animal tested for ECC filtration as stated in line 137. From the statistical point of view,

one animal is not enough to present even it is for the proof of concept research.

Response: The text in line 137 the reviewer referred to was: “In one animal, ECC filtration was

performed inside a fluorescence imaging system, and an imaging module was added to the

circuit to enable real-time fluorescence monitoring of blood (Fig. 4a).” We wanted to say here,

that we performed a study in only one animal inside the imaging system. We apologize for this

confusion. We performed ECC filtration in 3 animals/group with, and without ECC filtration for

the data presented in Fig. 4 (old Fig. 3). The study inside the imaging system was simply to

confirm the ability to monitor amount of drug removed in real-time. In the studies outside the

imaging system, amount of drug removed was determined based on blood samples. I.e., the same

data as in old Fig. 4 (new Fig. 5) was also acquired in studies for old Fig. 3 (new Fig. 4).

Therefore, the study in the imaging system does not provide any new additional data regarding

the ECC filtration system performance compared to old Fig. 3 (new Fig. 4). Furthermore, old

Fig. 3 (new Fig. 4) data is likely more accurate since concentration was determined from

extracted blood samples rather than imaging. The real-time monitoring is relevant for potential

use in human patients as described in the discussion, but it has limited relevance for the current

animal study. The most important data are shown in new Fig. 4 (old Fig. 3), where the ability to

remove drug by ECC filtration is established and amount of drug removed was calculated (see

section 2.2).

Comment #5: Figure 5 is not explained enough. No quantified data or the number of

measurements is shown. If it is based on the imaging only one animal, the imaging shall be

repeated to get some statistical comparison.

Response: We added a plot to Fig. 5 (new Fig. 6) that shows the mean tumor fluorescence in 4

animals per group, together with statistical comparison.

Reviewer 3 Report

Dear Author,

the manuscript "Extracorporeal Removal of Thermosensitive Liposomal Doxorubicin from Systemic Circulation after Tumor Delivery to Reduce Toxicities: Proof of Concept Study" focuses demonstrating a method for removal of systemic TSL-Dox 19 remaining after hyperthermia-triggered delivery. Also, the study is working on a relevant topic the novelty is not presented clearly. This needs to be improved more. Similar systems can be found by searching for them. It would be helpful to have more in vitro experiments along with the in vivo experiments.

Further the optical appearance of the graphs has to be improved. I suggest to us prism or else.

Additionally, I think more statistical analysis is required.

Author Response

Reviewer #3:
Dear Author,
the manuscript "Extracorporeal Removal of Thermosensitive Liposomal Doxorubicin from Systemic Circulation after Tumor Delivery to Reduce Toxicities: Proof of Concept Study" focuses demonstrating a method for removal of systemic TSL-Dox remaining after hyperthermia-triggered delivery.

Comment #1: Also, the study is working on a relevant topic the novelty is not presented clearly. This needs to be improved more. Similar systems can be found by searching for them. It would be helpful to have more in vitro experiments along with the in vivo experiments.

Response: We agree with the reviewer that multiple methods for removal of liposomal drug from systemic circulation have been described in prior studies. Unfortunately, those methods take several hours and are not sufficiently fast for TSL, since TSL-encapsulated doxorubicin have a half-life of ~1-2 h. The novelty of the proposed method is that we first release the drug, and that
removal of released drug can be performed much faster than removal of liposomes (e.g., in a single pass through an activated carbon filter, as we show in the current study). In addition, this method is only applicable to TSL based in intravascular triggered release, because there the delivery occurs only during heating (see Fig. S1). Therefore, removal of drug after heating reduces toxicity without affecting efficacy.

Comment #2: Further the optical appearance of the graphs has to be improved. I suggest to use prism or else.

Response: We re-created bar graphs with prism as suggested by the reviewer to improve appearance.

Comment #3: Additionally, I think more statistical analysis is required.

Response: We added additional statistical analyses, and Fig. 3b, Fig 4a, Fig 4d, and Fig 6b include now statistical evaluations.